# A Comparison of the Composition of Selected Commercial Sandalwood Oils with the International Standard

**DOI:** 10.3390/molecules26082249

**Published:** 2021-04-13

**Authors:** Malgorzata Kucharska, Barbara Frydrych, Wiktor Wesolowski, Jadwiga A. Szymanska, Anna Kilanowicz

**Affiliations:** 1Department of Toxicology, Faculty of Pharmacy, Medical University of Lodz, 90-151 Lodz, Poland; barbara.frydrych@umed.lodz.pl (B.F.); anna.kilanowicz@umed.lodz.pl (A.K.); 2Department of Biological and Environmental Monitoring, Nofer Institute of Occupational Medicine, 91-348 Lodz, Poland; wiktor.wesolowski@imp.lodz.pl; 3Retired Professor of Toxicology Department, Medical University of Lodz, 90-151 Lodz, Poland; jadwiga.szymanska@umed.lodz.pl

**Keywords:** essential oils, GC–MS, *Santalum album*, *Amyris balsamifera*, synthetic scents, adulteration

## Abstract

Sandalwood oils are highly desired but expensive, and hence many counterfeit oils are sold in high street shops. The study aimed to determine the content of oils sold under the name sandalwood oil and then compare their chromatographic profile and α- and β santalol content with the requirements of ISO 3518:2002. Gas chromatography with mass spectrometry analysis found that none of the six tested “sandalwood” oils met the ISO standard, especially in terms of α-santalol content. Only one sample was found to contain both α- and β-santalol, characteristic of *Santalum album*. In three samples, valerianol, elemol, eudesmol isomers, and caryophyllene dominated, indicating the presence of *Amyris balsamifera* oil. Another two oil samples were found to be synthetic mixtures: benzyl benzoate predominating in one, and synthetic alcohols, such as javanol, polysantol and ebanol, in the other. The product label only gave correct information in three cases: one sample containing *Santalum album* oil and two samples containing *Amyris balsamifera* oil. The synthetic samples described as 100% natural essential oil from sandalwood are particularly dangerous and misleading to the consumer. Moreover, the toxicological properties of javanol, polysantol and ebanol, for example, are unknown.

## 1. Introduction

Essential oils (EO) have accompanied people for centuries. Recent years have seen a return to natural remedies, including EO, in cosmetics, personal protection products, food, aromatherapy and medicinal purposes. Research conducted in recent years has found essential oil components to have a wide range of pharmacological effects, including antimicrobial, antioxidant and relaxing properties [1,2,3]. EO’s chemical composition, and thus the pharmacological and toxicological properties, depends on the species or plant part (leaves, flowers, bark, roots, etc.) from which it is obtained.

Such wide use of EO entails the need for effective quality control. In Europe, when used as food additives, they must comply with the requirements of the European Food Safety Authority (EFSA), and when used as cosmetics, they must comply with EU Regulation 1223/2009 on cosmetic products. In some cases, detailed standards have been developed regarding the quality of selected EO. For example, sandalwood oils should meet international standard ISO 3518:2002, which specifies requirements for its physicochemical properties, such as appearance, color, odor and density, as well as its chromatographic profile [4,5].

Among the many essential oils available on the market, sandalwood oil is one of the most sought-after, particularly for its healing properties. Sandalwood oil is one of the most expensive volatile oils available on the market and one of the most commonly faked. Sandalwood and its products have been known for their healing properties since ancient times, both in traditional Chinese medicine and in Ayurveda [2,6].

The sandalwood tree can be found in only a few regions globally, including South East Asia, Australia and some islands in the Pacific Ocean. The most precious species, *Santalum album* L., grows in India, the largest sandalwood oil manufacturer. The oil can also be produced from *Santalum spicatum*, an Australian species, and *Santalum ellipticum*, a Hawaiian species; however, these species are not regarded as possessing such valuable and therapeutic properties as the original oil extracted from *Santalum album*, mainly due to their lower content of α-santanol, believed to be the primary active component [7,8,9].

The most valuable variety of sandalwood oil from India (Oleum *Santalum album*) is believed to have exceptional organoleptic and technological properties, making it an exceptional perfume component. It possesses a subtle, characteristic and very durable fragrance and displays excellent properties for achieving long-lasting and harmonious scents. The oil also possesses a wide spectrum of antiseptic, anti-inflammatory, and diuretic effects and has been found to have a positive impact on skin conditions [10,11]. Recent studies also suggest that sandalwood oil may have anti-carcinogenic effects [12]. It is regarded as a safe aromatic food additive and has been accepted for use by the European Council and the American Federal Department of Drugs and Agriculture (FDA) [13]. The safety of sandalwood oil and its main component α‑santalol has been confirmed by toxicological data: its LD50 has been found to be greater than 2 g/kg of body weight, irrespective of the route of administration and animal species [5].

Oils with a similar smell can be obtained from other plant species, one of which is oleum *Amyris balsamifera*, from western India. This Western India sandalwood oil is produced on large *Amyris balsamifera* plantations and is less expensive than that obtained from *Santalum album* L. This *Amyris balsamifera* oil is an excellent natural mio-relaxant, i.e., a substance, which relaxes skeletal muscles due to the presence of high amounts of valerianol [1,14,15].

Sandalwood oils may also be faked by adding synthetic and semi-synthetic compounds. However, such adulteration affects their therapeutic properties by altering their chemical and physical properties; for example, it can increase their allergenic effect.

To ensure the quality of the sandalwood oils available on the market, various authors have suggested using advanced analytical methods [7,13,16]. According to ISO 3518:2002 standard, sandalwood oil (*Santalum album*) should be composed of not less than 41–54% of α-santalol and 16–24% of β-santalol [4,7]: sandalwood oils with lower levels are considered inferior quality. It has been proposed that the authenticity of these essential oils may be confirmed most effectively by a GC/MS approach, i.e., gas chromatography with mass spectrometry [17].

This study aimed to determine the content of volatile components of oils sold under the name *sandalwood oil*. It also compares the chromatographic profile of the oils and their content of sesquiterpene alcohols, especially α- and β-santalol, with the requirements of ISO 3518:2002. The tested oils were obtained from various producers, and they varied concerning price and source of origin, as declared by the producers.

## 2. Results

The results are presented in Figure 1 and in Table 1 and Table 2. The assessed oils generally displayed different chromatographic profiles (Figure 1). The percentage shares of selected groups of compounds in the investigated oils are presented in Table 1, while the shares of substances with the highest chromatographic peaks, or those characteristics of the specific oil, are given in Table 2.

Despite being manufactured by different companies, oils P1 and P2 did not differ qualitatively or quantitatively. In both, the main peak corresponded to valerianol (Figure 2), sesquiterpene alcohol with retention time (RT) of 22.19 min; this appeared to constitute 14.5% of the total content (Table 2). Sesquiterpene alcohols (general molecular formula: C_15_H_26_O, Figure 2) constituted more than half of the content (P1 55.2% and P2 53.8%) (Table 1). In addition to valerianol, different isomers of eudesmol (~26%) and elemol (~9%) produced significant peaks (Table 2, Figure 2). Approximately 27% of the content in P1 and P2 was made up of sesquiterpene hydrocarbons, with the general formula C_15_H_24_ (Figure 2); of these, 7-epi-α-selinene (~9%) and α-curcumene (~6%) were prevalent.

Oil P3 appeared to be qualitatively similar to P1 and P2. However, valerianol constituted only 8% of the total content, while caryophyllene made up 11.7% (RT 17.65 min). The latter was the most predominant compound, despite being present in minute quantities in P1 and P2 (Table 2). Compared to P1 and P2, P3 demonstrated higher levels of sesquiterpenes (C_15_H_24_), such as caryophyllene, cedrene and thujopsene (~44%) and lower levels of sesquiterpene alcohols (<40%) (Table 1). Apart from valerianol, mentioned before, eudesmol (~16%) and elemol (6%) (Table 2, Figure 2) isomers were detected in much less significant quantities.

In contrast, in the P4 spectra, sesquiterpene alcohols clearly predominated (49%), while the compounds detected in the previous oils (Table 1) were present in minute quantities. Of these, high amounts of polysantol (9.2%) and javanol isomers (24.4%) were observed (Table 2, Figure 3). Eudesmol isomers constituted only 6.1%, elemol 3.2%, and valerianol 4.7%. In addition, ebanol was present in two peaks, constituting 8.4% in total (Table 2, Figure 3). Compared to P1–P3, a lower level of sesquiterpenes (C_15_H_24_) was recorded (12.9%), such as selinene (2.6%), curcumene (1.5%) and caryophyllene (0.9%). Unlike P1–P3, cadinene (1.5%) and murolene (1.4%) were also found to be present. In addition, linalool (3.6%), linalyl acetate (3.9%) and farnesyl acetate isomers (3.2%) were unique to this oil (P4).

Sample P5 mostly contained benzyl benzoate (RT 28.24 min), which constituted 74.6% of the chromatogram (Table 2, Figure 4). Sesquiterpenes constituted 2% (thujopsene and cedrene) and sesquiterpene alcohols 1.9% (cedrol and bacdanol) (Table 2). The remaining significant peaks were identified as isopropyl myristate (bisomel)—2.0% (Figure 4), piperonal—1.4%, 2,2′-oxo(bis-1-propanol)—0.8%, vanillin—0.8%, and two unidentified peaks constituting 1.2% and 2.0%. All detected substances, apart from cedrol, thujopsene and cedrene, were absent from the other oils (Table 2).

P6 differed from the other oil samples. The predominant compounds were α- and β-santalol (Figure 5), which constituted 37.1% and 22.0% of the total content, respectively, with retention times (RT) of 23.38 min and 24.37 min (Table 2). Additionally, between 23.87 and 25.0 min, three more peaks appeared; these were identified as other santalol isomers, together approximating 7.9% of the content. At 23.42 min, a bergamotenol (Figure 5) peak was detected (6.7%), which co-eluted with α-santalol. In addition, lanceol (1.9%) and (*Z*)-nuciferol (1.1%) were observed. Sesquiterpene alcohols amounted to 78.3% of the compounds detected in this oil. It should be emphasized that these were completely different alcohols (with molecular formula C_15_H_24_O) from those in the other samples (Table 1); no valerianol, eudesmol or elemol isomers were detected. A unique sesquiterpene profile was identified: sesquiterpenes constituted 9.1% of the total content, and these included a range of different santalene isomers (5.9%) and α-himachalen (3.2%) (Table 2), which were not detected in other examined oils.

**Table 2 molecules-26-02249-t002:** Selected compounds detected in the examined “sandalwood” oils.

RT (min)	Compound	CAS	Area (%)						Molecular Formula
			P1	P2	P3	P4	P5	P6	
**Sesquiterpene hydrocarbons**									
17.65	Caryophyllene	87–44-5	0.15	0.15	11.70	0.93			C_15_H_24_
18.10	β-Himachalene	1461-03-6	2.15	2.24	1.47	0.42			C_15_H_24_
18.83	α-Curcumene	644-30-4	5.94	6.21	4.95	1.52			C_15_H_24_
21.98	α-Himachalene	3853-83-6						3.20	C_15_H_24_
22.37	7-epi-α-Selinene	123123-37-5	8.73	8.58	4.33	2.60			C_15_H_24_
**Sesquiterpene alcohols**									
19.50 19.70	Ebanol	67801-20-1				4.63 3.78			C_14_H_24_O
19.67	Polysantol	107898-54-4				9.21			C_15_H_26_O
20.93	Elemol	639-99-6	9.15	9.42	6.00	3.15			C_15_H_26_O
21.39	Bacdanol	28219-61-6					0.78		C_14_H_24_O
21.37	τ-Eudesmol	1209-71-8	6.88	6.97	3.72	2.16			C_15_H_26_O
21.56	Cedrol	16230-29-8	0.96	0.96	5.99		1.08		C_15_H_26_O
21.63 21.73	Javanol	198404-98-7				10.86 13.56			C_15_H_26_O
21.75	10-epi-γ-eudesmol	15051-81-7	5.87	5.98	3.82				C_15_H_26_O
22.19	Valerianol	20489-45-6	14.53	14.38	8.04	4.72			C_15_H_26_O
22.29	α-Eudesmol	473-16-5	4.98	4.95	3.17	1.41			C_15_H_26_O
22.41	β-Eudesmol (β-selinenol)	473-15-4	7.94	7.89	5.42	2.49			C_15_H_26_O
23.38	α-Santalol	115-71-9						37.12	C_15_H_24_O
23.42	Bergamotenol	88034-74-6						6.71	C_15_H_24_O
23.87	(*Z*)-α-Santalol	98718-52-6						0.83	C_15_H_24_O
24.18	*E*-*cis*,epi-β-Santalol	14490-17-6						4.45	C_15_H_24_O
24.37	β-Santalol	77-42-9						21.95	C_15_H_24_O
24.97	*trans*-β-Santalol	98718-53-7						2.64	C_15_H_24_O
25.14	*cis*-Lanceol	10067-28-4						1.85	C_15_H_24_O
25.65	(*Z*)-Nuciferol	78339-53-4						1.12	C_15_H_22_O
**Other substances**									
16.94	Linalool	78-70-6				3.55			C_10_H_18_O
17.08	Linalyl acetate (bergamiol)	115-95-7				3.88			C_12_H_20_O_2_
20.38	Isopropyl myristate (bisomel)	110-27-0					2.00		C_17_H_34_O_2_
21.92 22.24	Farnesyl acetate	29548-30-9/4128-17-0				1.98 1.24			C_17_H_28_O_2_
28.24	Benzyl benzoate	120-51-4					74.60	0.58	C_14_H_12_O_2_

RT—retention time (min).

## 3. Discussion

Sandalwood oil is a sought-after product due to its unique fragrance, price and pharmacological properties. The anti-inflammatory, anti-infective and antiproliferative effects of sandalwood oil (*Santalum album* oil, SAO), and its main ingredient—α-santalol, have been demonstrated in many preclinical studies [11,18]. The anti-inflammatory effects of SAO are believed to be mediated inter alia by inhibition of the cyclooxygenase 1 and 2 and 12-lipoxygenase pathways [19].

*Santalum album* oil also has been found to be active against many Gram-positive strains of bacteria, including *Staphylococcus*, including strains resistant to MRSA and VRSA antibiotics, *Streptococcus* and some Gram-negative bacteria [6]. In studies by Kačániová et al. [8], *Amyris balsamifera* oil exhibited strong antibacterial activity, and SAO possessed a moderate inhibition effect against *Staphylococcus* spp.

Natural sandalwood oil and its main ingredient, α-santalol, have been reported to have chemopreventive effects on skin cancer [20]. Research indicates that SAO can arrest the cell cycle at G2/M; it also can induce cell death via apoptosis. SAO also has been shown to induce autophagy and cell death in proliferating keratinocytes, suggesting that it may prevent the development of precancerous conditions [21].

The above-described effect of sandalwood oil is attributed either to its main component (α‑santalol) or to the entire mixture of active substances contained in this oil. Therefore, it is important that the oil used for medicinal purposes has a composition known to elicit the effects described above. However, the quality of the available oils, even at the source, i.e., in the country of origin, leaves much to be desired: of the 28 commercially available sandalwood oils produced in various provinces of India, less than 1/3 met the requirements of the India standard IS 329–2004 [22,23]. The requirements of this standard are similar to those of ISO 3518:2002, which indicates that the acceptable range of α-santalol content is 41–55% and β-santalol 16–27% [4]. However, the tested samples by Bisht’a et al. [23] were characterized by α-santalol levels from 0 to 54.28% and β-santalol from 0 to 25.94%, no santalols were found at all in 15 samples. The main adulterants were diethyl phthalate and diethylhexyl phthalate, which constituted up to 44% and 80%, respectively, in some of the commercial samples. The chromatographic profile of many samples also differed from the standard’s requirements and the chromatograms of reference samples created by the authors [23].

The safety of sandalwood oil is an important consideration. While routine allergy studies have found that only 0.1 to 2.4% of tested patients were allergic to the oil, many did not report the quality, purity or origin of the oil, which could have influenced the test results. For example, West Australian (*Santalum spicatum*) or Hawaiian (*Santalum paniculatum*) oil contains significant amounts of farnesol [11], which is listed as an irritant in Annex III of the EU Cosmetics Regulation [24]. However, this is not present in *Santalum album* oil.

The “sandalwood oils” examined in the present study may have completely different profiles, and hence, different biological properties to authentic *Santalum album* oil. Our data indicate that, while the P1, P2 and P3 oils differed quantitatively, they have similar qualitative content: all three samples contained valerianol and various eudesmol isomers, components characteristic of *Amyris balsamifera*. With this in mind, oils P1 and P2 were correctly labeled, while P3, labeled “*Santalum album* 100% bionatural”, was not: it did not contain any components characteristic of *Santalum album* oil. Sesquiterpene alcohols, such as valerianol and eudesmol, appear to be the main components of *Amyris balsamifera* oils [14,25]. These constituents are believed to be responsible for their relaxing properties.

The main components of *Santalum album* oil are α- and β-santalol, which constitute 57–89% of total content [7]. These were found to be present in oil P6, described by the manufacturer as “natural essential sandalwood oil from India”. In this product, α-santalol constituted 37.1% of total content, β-santalol 22.0% (Table 2 and Table 3), and 78.3% sesquiterpene alcohols (Table 1).

The oil examined in our study had an α-santalol content slightly lower than literature values of *Santalum album* oils (Table 3) but higher than the oil obtained from *Santalum spicatum*. The β-santalol content approximated that determined by Kačániová et al. [3], Panto et al. [26] and Góra and Gibka [27]. Our results confirm that the examined oil was obtained from *Santalum album*; however, it does not fulfill ISO 3518:2002 standard [4] due to an insufficient α-santalol content (<41%).

The main component of P5 was benzyl benzoate (74.6%) (Table 2), which is commonly used in the chemical, cosmetic and pharmaceutical industries [29]. This substance can be detected in natural conditions, being present in some plants, but it can also be obtained by chemical synthesis [30]. Other P5 components, such as isopropyl myristate (bisomel) or bacdanol (sesquiterpene alcohol), are also synthetic compounds and have a scent resembling that of sandalwood fragrance [30]. In addition, the thujopsene, cedrene and cedrol observed in P5 were also detected in other investigated samples, particularly those obtained from *Amyris balsamifera*. Hence, P5 appears to be a mixture of natural and synthetic components, which imitate the scent of sandalwood oil. No origin is given on the packaging; however, the description contains information about the presence of allergenic ingredients, following EU and EU Council Act no. 1223/2009 [24], concerning cosmetic products. Benzyl benzoate is placed in the top position in the ingredient list as the most important mixture component.

In addition, oil P4 also was found to have unique content. A comparison of its mass spectra with the FFNSC3 database enabled us to identify two javanol isomers with identical mass spectra, two ebanol isomers and polysantol. All of these compounds are synthetic substitutes, imitating the scent of sandalwood [17,30]. Although such synthetic oils are easy to distinguish from natural ones using GC–MS, it is difficult to confirm the identity of individual peaks due to an absence of standard mass spectra in commonly accessed databases, such as NIST or Wiley. While some matches can be found in fragrance mass spectra databases, such as FFNSC3, it is impossible to distinguish isomers of components with the same mass spectra (e.g., javanol I and javanol II).

Additionally, P4 was found to contain compounds, which can be obtained from *Amyris balsamifera* oil (valerianol, elemol and eudesmol isomers), but in far smaller quantities than in the pure oil. Although this oil, according to the description, was supposed to come from the *Santalum album*, α- and β-santalol were not detected in its composition. This suggests that P4 is a synthetic composition imitating sandalwood fragrance, not “natural essential oil 100%” from the sandalwood tree.

Fragrant substances have been used for centuries and have played warning, relaxing and healing roles, among others. Yet, in recent years, more attention has been paid to the safety of such substances stemming from their widespread and common use in the so-called “personal care products”, detergents and food, which might result in possible accumulation, leading to allergic reactions, dermatosis (contact dermatitis), neurological and respiratory problems [31]. The European Parliament and EU Council Act (WE) no. 1223/2009 concerning cosmetic products contains a list of allergenic substances, which may be used in cosmetics, but information about their presence must be included on the packaging [24]. These substances include various volatile oil components detected in the examined samples, such as d-limonene, geraniol, benzyl benzoate and linalool. Only in the case of one oil, sample P5 was such information provided.

Recent studies also indicate that fragrant substances may have an effect on the hormonal system [32]. Pick et al. [33] demonstrated that the main constituents of P4, viz. javanol and polysantol, act as estrogen receptor agonists (αER). Although javanol has much less estrogenic activity than estradiol, it may have synergic effects in conjunction with other components of fragrant substances.

## 4. Materials and methods

### 4.1. Materials

Six oils, sold as “sandalwood oils”, were selected for examination (Table 4). Two of them were described by the manufacturers as being obtained from *Amyris balsamifera*, three as original *Santalum album* and one without any provided source of origin.

Samples for the tests contained 1% solutions of the subsequent oils in dichloromethane (Avantor Performance Materials Poland S.A., Gliwice, Poland, purity 99.8%).

### 4.2. Methods

The assays were performed using an Agilent Technologies 6890N gas chromatograph (Wilmington, DE, USA), combined with an MSD 5973 mass spectrometer and split/splitless injection chamber with an HP-INNOWax polar capillary column (60 m × 0.25 mm i.d. × 0.5 µm d_f_). The chromatographic separation conditions were set to separate the peaks of the analyzed components from each other and dichloromethane. The chromatographic oven operated in the temperature range 50–240 °C based on the following programmable conditions: 50° for 2 min, ramping by 5 °C/min up to 80 °C. Then, the temperature ramped by 10 °C/min up to 120 °C and then finally by 20 °C/min to 240 °C. The carrier gas was helium, administered at a flow rate of 1 mL/min, and 1 µL injections (split 20:1) at 250 °C were made by autosampler. The leak from the chromatographic column was analyzed after electron ionization (EI), whereas the obtained positive ions were registered in scan mode within the range 10–350 Th. The obtained mass spectra were compared with standards given in the NIST11 or FFNSC3 spectrum databases (Mass Spectra of Flavors and Fragrances of Natural and Synthetic Compounds, 3rd Edition).

## 5. Conclusions

Among the six “sandalwood oils” tested, none of them met the requirements of ISO 3518:2002 concerning the content of the main component (α-santanol). By assessing the chromatographic profile (chemical composition) of the samples, it can be concluded that:Only one oil (P6) turned out to be the genuine sandalwood oil—*Santalum album*, and the description corresponded to the information provided by the manufacturer;Three samples (P1, P2 and P3) were obtained from *Amyris balsamifera*; two manufacturers provided correct information (P1 and P2) about its origin, while the third incorrectly claimed it to be *Santalum album*;Two oils (P4 and P5) were synthetic mixtures, which were supposed to imitate sandalwood fragrance. While P4 was labeled as being obtained from *Santalum album*, it turned out to be a mixture of chemical substances synthesized under laboratory conditions. The manufacturer of P5 did not indicate any source of origin.

The information provided on the oil packaging may help eliminate the oils of unknown origin or different origin than *Santalum album*. Unfortunately, such descriptions are often incomplete or misleading. The oil price is not a suitable criterion, either, although when purchasing a sandalwood oil at a price similar to those of lavender or pine oil, the customer should not expect a good quality sandalwood oil from *Santalum album*.

Our study is the first description of various “sandalwood oils” available in stationary and online stores, some of which contain mostly synthetic compounds. One oil contained benzyl benzoate primarily, and another was composed of mostly sesquiterpene alcohols, such as javanol, ebanol and polysantol, which are supposed to imitate the fragrance of genuine sandalwood.

Sandalwood oil is highly appreciated, not only because of its exotic fragrance connotations and intensity but also because of its healing properties, which can be used in aromatherapy, for example. Natural oils, even those not derived from sandalwood, do not appear to pose a serious health risk. On the other hand, oils from plants other than the *Santalum album* will not have pharmacological effects appropriate for SAO. However, it should be emphasized that synthetic oils may contain compounds of not fully recognized biological properties, and which may be hazardous for the health of the user.

## Figures and Tables

**Figure 1 molecules-26-02249-f001:**
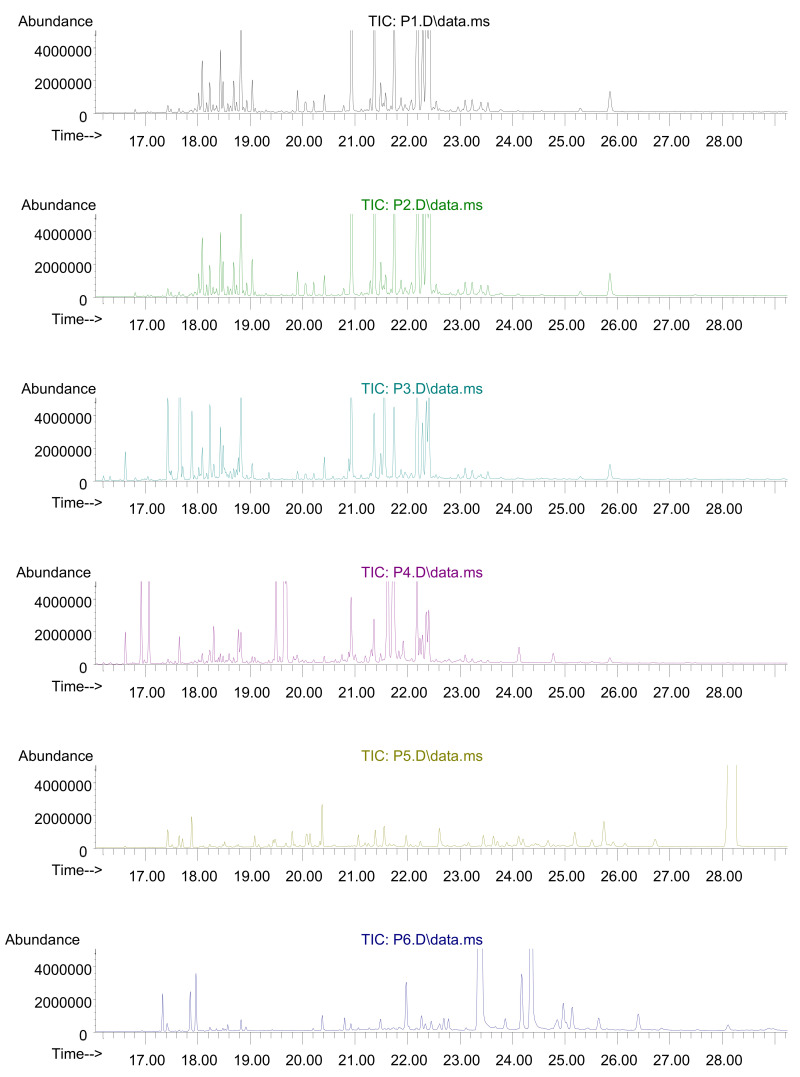
Chromatograms of the assessed sandalwood oil samples (explanations of P1–P6 are included in Table 4).

**Figure 2 molecules-26-02249-f002:**
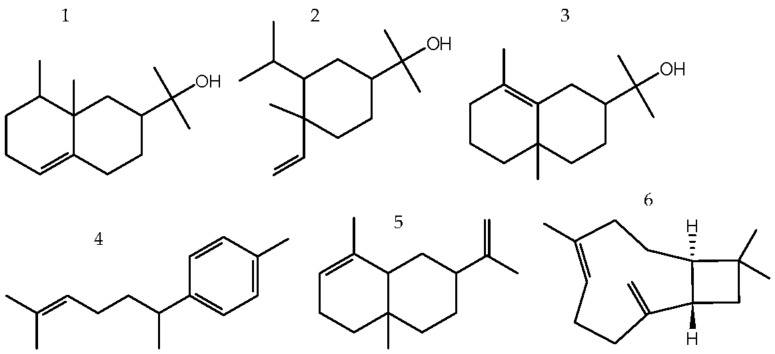
Structural formulas of selected compounds detected in the examined “sandalwood” oils. Main components in samples P1, P2, P3—sesquiterpene alcohols: 1—valerianol, 2—elemol, 3—eudesmol; sesquiterpene hydrocarbons: 4—α-curcumene, 5–7-epi-α-selinene, 6—β-caryophyllene.

**Figure 3 molecules-26-02249-f003:**
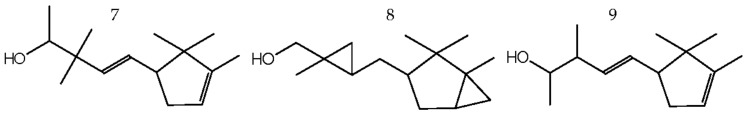
Structural formulas of selected compounds detected in the examined “sandalwood” oils. Main components in sample P4—sesquiterpene alcohols: 7—polysantol, 8—javanol, 9—ebanol.

**Figure 4 molecules-26-02249-f004:**
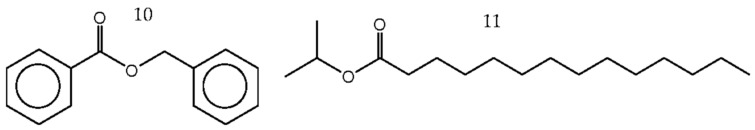
Structural formulas of selected compounds detected in the examined “sandalwood” oils. Main components in sample P5—other substances (esters): 10—benzyl benzoate, 11—bisomel.

**Figure 5 molecules-26-02249-f005:**
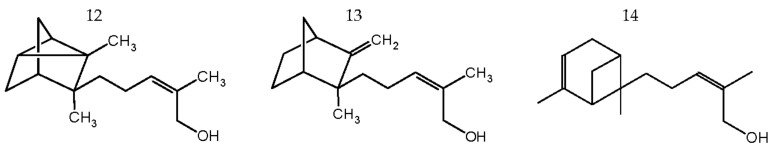
Structural formulas of selected compounds detected in the examined “sandalwood” oils. Main components in sample P6—sesquiterpene alcohols: 12—α-santalol, 13—β-santalol, 14—bergamotenol.

**Table 1 molecules-26-02249-t001:** The percentage shares of selected groups of substances in the assessed samples.

Group of Substances (Molecular Formula)	Percentage Share of Selected Groups of Substances					
	P1 *	P2 *	P3 *	P4 *	P5 *	P6 *
Sesquiterpene hydrocarbons C_15_H_24_	26.9	27.1	43.9	12.9	2.0	9.1
Sesquiterpene alcohols C_15_H_26_O	55.2	53.8	39.9	49.1	1.1	0.0
C_15_H_24_O	0.0	0.0	0.0	0.0	0.0	78.3
Other sesquiterpene alcohols	0.0	0.0	0.0	8.4	0.8	1.1
Other substances	3.3	1.0	0.4	13.5	82.8	1.5

* P1–P6—explanations in Table 4.

**Table 3 molecules-26-02249-t003:** The content of selected components characteristic of Santalum oils.

Substance	Content (%)					
	P6 (Kucharska et al. Present Study)	*Santalum album*			*Santalum spicatum*	
		Kačániová et al. [3]	Panto et al. [26]	Góra, Gibka [27]	Hołderna-Kędzia et al. [28]	Shellie et al. [16]
α-Santalol	37.1	59.0	44.6	46.2	32.1	22.0
β-Santalol	22.0	9.0	19.2	20.5	22.0 *	5.2
Bergamotol	6.7	–	5.9	4.7	5.3	–
Nuciferol	1.1	1.7	4.0	1.2	– *	2.8
Lanceol	1.9	1.9	1.4	1.5	–	5.2
Trans-farnesol	–	–	–	bd	4.1	5.8

* β-santalol + nuciferol.

**Table 4 molecules-26-02249-t004:** Description of the assessed oils.

Sample Symbol	Name of the Oil	Source of Origin	Content on the Packaging
P1	West India natural sandalwood oil	*Amyris balsamifera oil*	*Amyris balsamifera oil*
P2	Sandalwood oil	*Amyris balsamifera oil*	*Amyris balsamifera oil*
P3	Essential oil—“SANDAL”	*Santalum album* 100% bio-natural oil	–
P4	Sandal tree oil	*Santalum album *(*sandalwood oil*)** Natural essential oil 100%	–
P5	“Sandalwood” scented oil	–	Benzyl benzoate, coumarin, hydroxycitronellal, benzyl alcohol, benzyl salicylate, benzyl cinnamate, isoeugenol, limonene, eugenol
P6	Natural essential sandalwood oil from India	*Santalum album wood oil*	–

## Data Availability

Not applicable.

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
