# Peer review of "A Comparison of the Composition of Selected Commercial Sandalwood Oils with the International Standard"

_molecules, 2021, doi:10.3390/molecules26082249_

Round 1

Reviewer 1 Report

This work was addressed to analyze the composition of six different commercial “sandalwood oils”, also making a comparison between the oils themselves and the results from literature; as a result, three samples out of six resulted to be obtained from Amyris balsamifera, two were found having synthetical origin, and only one was actually obtained from Santalum album. As far as I can see, the work is new, original, English form is correct, scientific background is solid, conclusions are supported by the obtained results. Furthermore, in my opinion it may have a good potential impact on the related literature, representing an interesting starting point for detecting adulterations and commercial frauds in these commercial oils. In my opinion, the manuscript deserves to be published in the present form; I only found some minor points to be corrected, that I list hereby:

  • Introduction, line 45: as far as I know, the correct spelling should be “Ayurveda”.
  • Introduction, line 48: Santalum album is not a variety, rather a species. Rearrange the sentence to avoid misunderstandings. Similarly, at line 51 maybe discussing about “oil varieties” should be more proper.
  • Figure 1 and table 1: tables and figures should be self-explaining; check the opportunity to explain (in the caption, or in a footnote), what P1, P2 etc. are referred to.
  • line 177: write “of” with small letter
  • line 187: who are “those tested”? Do you mean “tested patients”?
  • Table 3: in the caption, write “Santalum” with initial in capital; in the body of the table, write correctly “Santalum spicatum”.
  • Lines 192-197: The “discussion” section should only discuss the results and their implications, on the basis of the available literature. I don’t think that the repetition of the aim if the work is useful here, since it has been already stated in other sections. I suggest to remove these lines.

Author Response

Responding to the comments contained in the review, we state that:

- changes were introduced in the text of the work in line with the Reviewer's suggestions, this applies to lines 45, 51, 48, 177, 187 and table 3;

- in the "Discussion", repetitions were removed (lines 192-197);

- descriptions of table 1 and figure 1 have been supplemented with the phrase: "explanations of P1-P6 are included in table 4 (" Materials ").

Thank you for evaluating our work.

Reviewer 2 Report

This manuscript was already submitted a few weeks ago. My main criticism was, and still is, that it does not bring any novelty in terms of scientific research. This is just the description of what any QC lab of the perfume industry would do to ensure the authenticity of sandalwood oils.

There is no point of writing a paper on that subject.

Author Response

The reappearance of our work in the journal is related to the invitation we received from the Editorial Board in connection with the preparation of a special issue dedicated to essential oils (Biomolecules from Essential Oil Bearing Plants: Biological and Industrial).

After editing and supplementing the publication, we decided that it corresponds to the profile of the issue being prepared.

We find it difficult to agree with the Reviewer's thesis that our research brings nothing new. There is little work on a similar topic in the scientific literature. The value of our work is confirmed by two positive reviews.

We are sorry that despite the changes we have made to our work, it was not approved by the Reviewer.

Reviewer 3 Report

This manuscript is useful as it reports on what is ‘out there’ in relation to the less expensive Sandalwood oils.  It seems that basically, in terms of quality,  ‘you get what you pay for’ is the rule that generally applies.

The samples have been analysed on a polar glc column, which is usually what is used for Sandalwood oil analyses and the glc traces of the given in the manuscript.  Putting the traces, in ‘portrait’ orientation on one page is really compressing the traces too much.  I would recommend that the glc traces be shown in ‘landscape’ mode and be done at 2 scans per page so that the readers can get a better idea of how the scans look.

Some of the compounds found in the fake sandalwood oils are synthetic compounds made by flavor companies, though their mass spectra appear to be contained in propriatry mass spec data bases.  This is a problem as any lab analysing these oils without this database would not know what to look for.  Are the mass spectra of  ‘polysantol’, the two ‘javanol isomers’ and the two ‘ebanol’ isomers published in the literature, to which readers cold be directed?

Table 2, ,lists the compounds identified in the oils in retention time order.  Linear Retention Indicex would be a much more useful identifier.  Did (or could) the authors include these indices in place of the retention times?

Table 3 , the name of the species is Santalum spicatum  not  Santalum spikatum.

Page 12 line 234.  Quote details of the FNSC# database.

A reference to an analysis of the oil from Amyris balsamifera (preferably on a polar column) would be useful.

The Reference section needs to be brought into line with that used by the Journal.

Author Response

Thank you for the positive evaluation of our work. Responding to the comments contained in the review, we would like to note that:

- the representation of all the chromatograms in one figure (Fig. 1) allows for a quick comparison of the tested samples. However, we have prepared a presentation of 3 chromatograms on one page, as suggested by the Reviewer, but we leave the choice of the version of Fig. 1 to the editorial office of the journal;

- in available works, including those cited by us, the mass spectra of javanol, ebanol and polysanthol are not available. They are also missing from general mass spectral libraries (eg, NIST 11). They are only available in spectral libraries dedicated to fragrances (such as eg FFNSC3);

- providing a retention index instead of the retention time is more useful for the identification of substances with a non-specific detector (eg FID). We believe that for our research it is more appropriate to include the retention time in Tab. 2. In our previous research we always used the retention time and identification of compounds confirmed by the mass spectrum;

- line 234 should be FFNSC3 (base name is explained on page 13, lines 275-276);

- the name of Santalum spicatum has been corrected;

- the analysis of the composition of Amyris balsamifera oil published in [14] was performed on a polar DB-WAX column, and the authors of [23] performed an analysis on a non-polar DB-5 column.

the literature (references) has been revised according to the rules in force in the journal.

Round 2

Reviewer 2 Report

Dear Editor,

I did my review for this manuscript and I still hold on my opinion, this is not a research paper.
Even if the authors were invited to contribute to a special issue, it does not prevent them from submitting something consistant in terms of science.

Now it's up to the Editorial Office to decide whether you would accept it or not.